# Prevalence and Spectrum of Second Primary Malignancies among People Living with HIV in the French Dat’AIDS Cohort

**DOI:** 10.3390/cancers14020401

**Published:** 2022-01-13

**Authors:** Isabelle Poizot-Martin, Caroline Lions, Cyrille Delpierre, Alain Makinson, Clotilde Allavena, Anne Fresard, Sylvie Brégigeon, Teresa Rojas Rojas, Pierre Delobel

**Affiliations:** 1APHM, Inserm, IRD, SESSTIM, Sciences Economiques & Sociales de la Santé & Traitement de l’Information Médicale, ISSPAM, APHM Sainte-Marguerite, Service D’Immuno-Hématologie Clinique, Aix Marseille Université, 13009 Marseille, France; 2APHM Sainte-Marguerite, Service D’Immuno-Hématologie Clinique, Aix Marseille Université, 13009 Marseille, France; caroline.lions@ap-hm.fr (C.L.); Sylvie.RONOT@ap-hm.fr (S.B.); teresa.rojas-rojas@ap-hm.fr (T.R.R.); 3Centre d’épidémiologie et de Recherche en Santé des Populations (CERPOP), Université de Toulouse, Inserm, UPS, 31100 Toulouse, France; cyrille.delpierre@inserm.fr; 4Département des Maladies Infectieuses et Tropicales, INSERM U1175/IRD UMI 233, Centre Hospitalier Universitaire de Montpellier, 34295 Montpellier, France; a-makinson@chu-montpellier.fr; 5Service des Maladies Infectieuses et Tropicales, CHU Hôtel-Dieu, 44000 Nantes, France; clotilde.allavena@chu-nantes.fr; 6Département des Maladies Infectieuses et Tropicales, Centre Hospitalier Universitaire de Saint-Étienne, 42270 Saint-Etienne, France; anne.fresard@chu-st-etienne.fr; 7Service des Maladies Infectieuses et Tropicales, Inserm, UMR 1291, Université Toulouse III Paul Sabatier, CHU de Toulouse, 31062 Toulouse, France; delobel.p@chu-toulouse.fr

**Keywords:** cancer survivors, HIV, AIDS, secondary primary cancer, AIDS-defining cancers, non AIDS-defining cancers

## Abstract

**Simple Summary:**

People who survive primary cancers are at an increased risk for subsequent primary cancers. An increased risk for certain types of primary cancers among people living with HIV (PLWH) was demonstrated in the last few decades. Given the increasing life expectancy of PLWH, a steady increase in SPC has been reported. The main objective of this study was to describe the prevalence and spectrum of second primary cancers (SPCs) stratified by first primary cancers in HIV-positive men and women cancer survivors. We showed that the pattern of SPCs differs from that observed in the general population and according to sex. Yet, further studies are needed to determine the excess risk of SPCs in this population and to confirm the need for more appropriate screening procedures.

**Abstract:**

Background: We aimed to describe the prevalence and spectrum of second primary cancer (SPC) in HIV-positive cancer survivors. Methods: A multicenter retrospective study was performed using longitudinal data from the French Dat’AIDS cohort. Subjects who had developed at least two primary cancers were selected. The spectrum of SPCs was stratified by the first primary cancer type and by sex. Results: Among the 44,642 patients in the Dat’AIDS cohort, 4855 were diagnosed with cancer between 1 December 1983 and 31 December 2015, of whom 444 (9.1%) developed at least two primary cancers. The most common SPCs in men were non-Hodgkin lymphoma (NHL) (22.8%), skin carcinoma (10%) and Kaposi sarcoma (KS) (8.4%), and in women the most common SPCs were breast cancer (16%), skin carcinoma (9.3%) and NHL (8%). The pattern of SPCs differed according to first primary cancer and by sex: in men, NHL was the most common SPC after primary KS and KS was the most common SPC after primary NHL; while in women, breast cancer was the most common SPC after primary NHL and primary breast cancer. Conclusion: The frequency and pattern of subsequent cancers among HIV-positive cancer survivors differed according to the first primary cancer type and sex.

## 1. Introduction

People who survive primary cancers are at an increased risk for subsequent primary cancers, and given the recent progress in the field of therapeutics in oncology, the risk of second primary cancer (SPC) could become a major concern in the near future [1,2,3,4]. In France, the risk of SPC is on average 36% higher among cancer survivors compared with the general French population [5]. In the United States, a 14% increased risk among cancer survivors compared with the general US population was reported [6], with a lifetime risk of developing a SPC estimated to be as high as 33% among selected subgroups of cancer survivors [7]. The type of the first cancer, age, primary cancer treatments, genetic susceptibility and environmental and lifestyle exposures, contributed to the occurrence of a SPC [8]. A clustering of SPCs was also identified for different cancer types that shared the same hormonal, genetic or lifestyle risk factors [8].

An increased risk for certain types of primary cancers among people living with HIV (PLWH) was demonstrated in the last few decades, although the level of risk has diminished for some cancers in recent years due to better control of viral replication and improvement in immune status [9,10,11]. Given the increasing life expectancy of PLWH, a steady increase in SPCs has been reported in the US [12]. The French CANCERVIH network reported an incidence of 13% of SPCs among cancer cases diagnosed in PLWH that were presented to the multidisciplinary staff from 2014 to 2019 [13].

We aimed to describe the prevalence and the spectrum of SPCs stratified by the type of first primary cancer in HIV-positive men and women in the French Dat’AIDS cohort.

## 2. Materials and Methods

This multicenter analysis was performed using longitudinal data from the French Dat’AIDS cohort (NCT 02898987, ClinicalTrials.gov), which represents a collaboration between 17 major French HIV clinical centers that use a common electronic medical record system (NADIS^®^) for the follow up of HIV-, hepatitis B virus (HBV)- and hepatitis C virus (HCV)-infected adults. The data collection was approved by the French National Commission on Informatics and Liberty (CNIL 2001/762876; Methodology Reference (MR) MR004/2210731v.0), and all patients signed an informed consent prior to inclusion in the cohort. For this study, the database was censored on the 31st of December 2015. We selected subjects who developed at least two primary cancers, which occurred after the date of HIV diagnosis.

### 2.1. Cancer Cases

Cancer cases were recorded by the clinicians in the database in each HIV clinical center. The International Statistical Classification of Diseases and Related Health Problems, Tenth Revision (ICD10, World Health Organization, Geneva) codes were used to identify cancer types (ICD codes C00 to C95). No time span between two cancer diagnoses was required, except for same cancer types for which a minimum delay of 5 years was mandatory to be considered as a SPC. Metastatic primary malignancies, secondary lymph nodes and primary cancer relapse were defined as the same cancer type occurring within 5 years and were thus excluded from analysis.

Cancers cases were classified into three categories: AIDS-defining cancers (ADCs), that is, Kaposi sarcoma (KS) (ICD10: C46), non-Hodgkin lymphoma (NHL) (ICD10: C82–C85) and invasive cervical cancer (ICC) (ICD10:C53); virus-related non-ADCs (VR-NADCs); and virus-unrelated-NADCs (VU-NADCs). VR-NADCs were cancers of the anus, vagina, vulva, penis and selected oral cavities or pharynx sites for potential human papilloma virus association (HPV) (ICD codes C01, C02, C09, C10, C14); liver cancer (C22) for HBV and HCV association, Hodgkin’s lymphoma (HL) (C81) for Epstein-Barr virus association; and Merkel cell carcinoma, associated with Merkel cell polyomavirus. VU-NADCs included all remaining cancers.

### 2.2. Data Collection

The data collected were sex; birth date (months, years); HIV transmission route (heterosexual, men who have sex with men (MSM), intravenous drug use (IVDU), others); year of HIV diagnosis; date of antiretroviral therapy (ART) initiation; and dates of each cancer event. Smoking and alcohol consumption were assessed at the last medical encounter and categorized as never, past or current. Regarding biological data, CD4 count nadir, HBV status (defined by a positive HBV surface antigen test) and HCV status (defined by HCV antibody positivity) were collected. HIV viral load (VL) and CD4 count at the time of each cancer event and at the time of HIV diagnosis were not available, neither were these data available at the time of treatment of the first cancer. Race and ethnicity were not accounted for by respect of French regulatory procedures.

### 2.3. Statistical Analysis

Primary cancer events were stratified by cancer categories and cancer types, and analyzed according to sex. The spectrum of SPCs was described overall and stratified according to the first primary cancer type and according to sex. The statistical analyses were performed using SAS statistical software (SAS Institute Inc., Cary, NC, USA, version 9.4).

## 3. Results

### 3.1. Patient Characteristics

Among the 44,642 PLWH in the Dat’AIDS cohort, 4855 were diagnosed with cancer between 1 December 1983 and 31 December 2015, of whom 444 (9.1%) developed at least two primary cancers. Out of the 444 patients with SPCs there were 397 (89.4%) with one SPC, 43 (9.7%) with two SPCs and 4 (0.9%) with three SPCs.

The studied population was composed of 369 men (83.1%) and 75 women (16.9%), with a median age at the first cancer event of 46 years and a median age at the SPC event of 51 years; 47.8% were MSM, 16.0% were coinfected with HCV, 9.1% were coinfected with HBV and 76.9% had a nadir CD4 count <200/mm^3^ (Table 1). The median follow-up time between the first primary cancer and the last medical encounter was 9 years (IQR: 4–15). Compared to men, women had more often acquired HIV through heterosexual intercourse and IVDU and were more often coinfected with HCV. They were also younger at the first cancer event, which occurred less frequently in the pre-ART period, occurred after a longer delay after ART initiation and belonged more frequently to the VU-NADC category.

One-third of patients declared that they were past or current smokers (31.5% and 37.3%, respectively) without a significant difference between men and women, while alcohol consumption differed by sex, with a higher rate of current consumers among men and never consumers among women.

### 3.2. Cancer Events According to the ADC, VR-NADC and VU-NADC Categories

The first primary cancer types included 269 ADCs, 51 VR-NADCs and 124 VU-NADCs (Table 1). The distribution of primary cancer categories differed by sex, with a higher proportion of VU-NADCs in women and a higher proportion of ADCs in men. Regarding SPC categories, the distribution was similar in men and women, and VU-NADCs were the most common in both sexes. The median delay between the first primary cancer and the SPC was four years, without a significant difference between men and women.

### 3.3. Spectrum of First Primary Cancer and SPC Types by Sex

SPCs for all primary cancers are reported in Appendix A and for those comprising more than five cases in Appendix A for women and Appendix A for men.

Among the 75 first primary cancers diagnosed in women, those that comprised more than five cases were NHL (17 cases, 22.6%), breast cancer (12 cases, 16.0%), cervical cancer (8 cases, 10.7%) and KS (7 cases, 9.3%) (Appendix A). Furthermore, cancers potentially related to HPV accounted for 16.0% of first primary cancer cases (12 cases, including 8 cases of cervical cancer, 2 cases of vulva cancer and 2 cases of anal cancer) (Table 2).

In men, first primary cancers comprising more than five cases, apart from skin carcinomas, were KS (149 cases, 40.4%), NHL (88 cases, 23.8%), HL (14 cases, 3.8%), anal cancer (12 cases, 3.2%), lung cancer (11 cases, 3.0%), prostate cancer (10 cases, 2.7%), kidney cancer (10 cases, 2.7%) and liver cancer (9 cases, 2.4%). Moreover, upper aerodigestive cancer (C00–C14, C30–C32) accounted for 5.1% of first primary cancers (Appendix A). Additionally, cancers potentially related to HPV accounted for 5.9% of the first primary cancer cases (Table 2).

Regarding SPCs, apart from breast and prostate SPC, the spectrum of the most common SPCs was similar for men and women, but their frequency differed by sex: NHL was the most frequent SPC in men (22.8%), and breast cancer was the most frequent SPC in women (16.0%) (Figure 1 and Figure 2).

### 3.4. Spectrum of SPCs According to the First Primary Cancer Type by Sex

In women, breast cancer was the most frequent SPC after NHL (C82–C85) (3 of 17 patients) and after a first breast cancer (4 of 12 patients). After cervical cancer, skin carcinoma was the most frequent SPC (3 of 8 patients). Moreover, when considering the 27 female cancers (identified by IC codes C50, C51, C53, C54, C55, C56 and C57), ten (37%) second primary female cancers occurred after a first primary female cancer, including breast cancer in seven patients, uterus cervix cancer in one patient, uterus corpus cancer in one patient and vulva cancer in one patient (Appendix A).

In men, KS was mainly followed by NHL (C82–C85), representing 31.5% of all SPC diagnoses after KS (47 of 149 patients) (Appendix A). Skin carcinoma, anal cancer and HL were the three other most common SPCs diagnosed after KS at 14.8 % (22 out of 149 patients), 8.7% (13 out of 149 patients) and 7.4% (11 out of 149 patients), respectively. Among survivors of NHL, KS was the most common SPC at 26.1% (23 out of 88 patients), followed by NHL, HL and lung cancer at 17.0% (15 out of 88 patients), 10.2% (9 out of 88 patients) and 6.8% (6 out of 88 patients), respectively. After primary HL, NHL was the most common SPC (7 out of 14 patients). Additionally, among the 19 survivors of upper aerodigestive cancers, cancers of the oral cavity were the most frequently diagnosed SPCs at 31.6% (6 out of 19 patients), followed by lung cancer at 19% (4 out of 19 patients).

Among the 34 subjects who had first primary cancers that were potentially related to HPV, including 12 cases in women and 22 cases in men, eight cases (23.5%) of second primary, potentially HPV-related cancer occurred, with two cases of anal cancer in women, and five cases of tongue cancer and one case of tonsil cancer in men (Table 2).

## 4. Discussion

In this large cohort of HIV-positive cancer survivors with a median follow-up time of 9 years since the first cancer event [4,5,6,7,8,9,10,11,12,13,14,15], the SPC prevalence was 9.1% (444/4855), confirming that SPCs are a major concern in this population.

This rate of prevalence was close to that observed in the general population. The Surveillance, Epidemiology and End Results Program (SEER) registries between 1975–2001 and between 1992–2008, indicated that in the general US population nearly 8% of cancer survivors had a history of more than one cancer [7,14]. In the general French population, data from several cancer registries covering 289 967 patients, with a first cancer diagnosed between 1989 and 2004 and follow up until 31 December 2007, indicated that 7.3% of patients developed a SPC [5].

The more severe prognosis in the pre-ART era for PLWH with cancer and the low percentage of subjects over 65 years of age in our cohort could explain the lower prevalence rate of SPCs observed in our cohort than expected, due to the high number of risk factors for cancer in PLWH.

This rate of prevalence was also consistent with the study on HIV-positive San Francisco residents conducted by Hessol et al., between 1985 and 2013, in which SPC accounted for 9% of all cancers [12] but was lower than reported by the French CANCERVIH network [13]. In this study, 13% of PLWH with a cancer diagnosed between 2014 and 2019 had a history of at least one cancer. This discrepancy might be explained by the fact that this study was conducted during a more recent period. Further studies are needed to confirm such evolution.

Regarding patients’ characteristics, as in the general population [15], SPCs occurred a few months or many years after the first cancer but at younger median ages than in the general French population [5,16,17], due to the differences between both populations in the age distribution [18].

Most subjects, both men and women, had a nadir CD4 < 200/mm^3^, but the proportion of those contaminated with HIV through IVDU and with HCV coinfection was higher in women than in men, potentially explaining younger ages in women at the first primary cancer [19]. The distribution of the first primary cancer categories, but not of the SPCs, differed by sex, with VU-NADCs being the most common SPCs. A higher prevalence of VR-NADCs might have been expected due to the high prevalence of coinfection with oncogenic viruses in PLWH. Data regarding co-infection with oncogenic viruses in our cohort were limited, making us unable to further explore this result.

Additionally, we observed that the delay between HIV diagnosis, first ART initiation and the first primary cancer were significantly longer in women than in men. The higher barriers to access and engagement in HIV care for women living with HIV might explain these disparities [20,21,22].

Regarding the pattern of first primary cancers in our cohort, the two most common first primary cancers were KS and NHL in men and NHL and breast cancer in women. In the general French population of cancer survivors, an excess risk of SPCs was identified after a first cancer of the head and neck, larynx, esophagus, bladder, lip, kidney, lung or chronic lymphatic leukemia in men, and after a first cancer of the larynx, head and neck, esophagus, vagina, vulva or acute myeloid leukemia in women [5]. Likewise, the spectrum of SPCs observed in our cohort also differed from that observed in the general French population [5], but it was consistent with the spectrum of primary cancers in the cohort, as reported [12,13,23,24].

Furthermore, the frequency and pattern of SPCs differed according to the first primary cancer, as reported by Hessol et al. [12], but also by sex.

In men, NHL was the most common SPC after primary KS, and KS was the most common SPC after primary NHL.

These results were consistent with previous studies that identified a higher risk of developing NHL after KS among PLWH than among the general U.S. population [23,25], and a higher risk of developing KS after an AIDS-defining lymphoid malignancy [24].

In women, breast cancer was the most common SPC after primary NHL, which was in agreement with data of Mahal et al. [24]. Additionally, as observed in the general population of French women [26], breast cancer was also the most common SPC after a primary breast cancer. This result is of major concern in France since breast cancer screening has been reported to be lower in women living with HIV (WLWH) than in the general population (82.2% versus 88%, respectively [27]). These data highlight the need to further encourage WLWH to be screened for breast cancer.

Furthermore, as reported in the general population, we observed a clustering of genital cancers in women with first primary cancers of the breast or the female reproductive tract [8].

Additionally, we found that 23.5% of SPCs that occurred after a first primary cancer that were potentially related to HPV in men and women, were also potentially related to HPV. This result was consistent with data from the general French population wherein after a potentially HPV-related first cancer, the second cancer occurred frequently at other HPV-related sites, in men and women [28]. In addition, long-term persistence of HPV, especially high-risk HPV, was more common in HIV-positive men and women, and may have contributed to the development of HPV-related SPCs in PLWH [29,30].

We observed that after primary cancer of the upper aerodigestive tract in men, cancers of the oral cavity and pharynx represented 41% of the SPCs, and lung cancer represented 17%. High levels of tobacco and alcohol exposure were related to the reciprocal associations observed between cancers of the head and neck, esophagus, larynx and lung in France [5], as in other countries [6,31,32,33]. Among the men in our cohort, 70.4% were past or current smokers and 57.1% were past or current alcohol consumers.

Our study had several limitations. First the rate of SPC reported herein may have been overestimated, given that these second malignancies could have been misclassified metastases or relapses of the first primary tumor. However, to limit this classification bias, all subjects with a cancer of the same type occurring within five years of the first cancer were excluded from the analysis. Furthermore, there is currently no common definition of second primary cancer; some authors consider all subsequent tumors, irrespective of the time lag between the first and second cancer diagnoses [34,35] while others consider those with at least a 6-month or 2-month delay [5,15]. Second, the potential for underreporting of cancers cannot be excluded. However, data were collected from an electronic medical record in which the medical staff were also responsible for data entry, thereby ensuring data quality and completeness. Third, we did not check the histologic report for each cancer case, and thus, we could not exclude the possibility of the misclassifications of high-grade lesions into cancer codes.

## 5. Conclusions

This descriptive analysis of the spectrum of SPCs in HIV-positive cancer survivors identified NHL in men and breast cancer in women as the two most common SPCs. However, clinicians must be aware that the frequency and pattern of SPCs differed according to the first primary cancer type and might change over time, as reported by Mukhtar et al. for SPCs associated with KS [36], arguing for regular epidemiologic monitoring. Further studies are needed to determine the excess risk of each SPC type according to the first primary cancer type in PLWH, and there is a need for more appropriate screening procedures in this population [37,38,39].

## Figures and Tables

**Figure 1 cancers-14-00401-f001:**
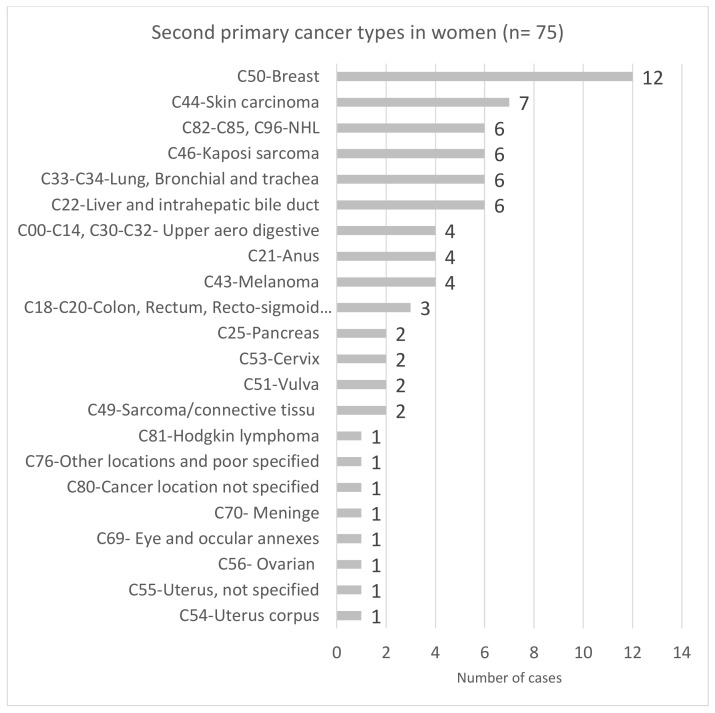
Spectrum of second primary cancer types in women living with HIV in the French Dat’AIDS cohort (*n* = 75). Abbreviation: NHL: non-Hodgkin lymphoma.

**Figure 2 cancers-14-00401-f002:**
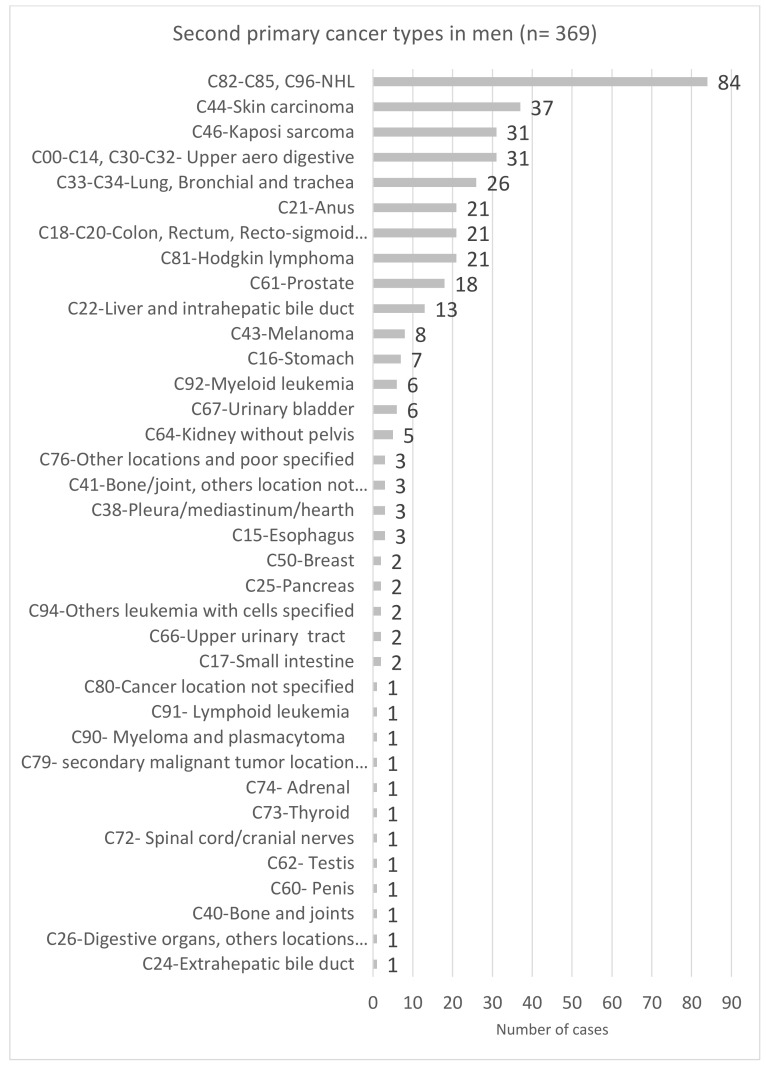
Spectrum of second primary cancer types in men living with HIV in the French Dat’AIDS cohort (*n* = 369).

**Table 1 cancers-14-00401-t001:** Characteristics of patients with a second primary cancer in the French Dat’AIDS cohort.

N (%); Median [IQR]	N = 444	Women	Men	*p*
		75 (16.89)	369 (83.11)	
Delay between HIV diagnosis and first primary cancer (y)	6.5 [1.0; 13]	9.4 [3.9; 14.9]	5.9 [0.4;12.3]	0.0004
Median follow-up time since first cancer event and the last medical encounter	9 [4; 15]	7 [3; 15]	9 [4; 15]	0.22
Age at first cancer (y)	46 [38; 54]	42 [36; 51]	47 [38; 55]	0.03
22–49 years	271 (61.04)	55 (73.33)	216 (58.54)	0.06
50–64 years	144 (32.43)	17 (22.67)	127 (34.42)	
65 years and over	29 (6.53)	3 (4.00)	26 (7.05)	
Age at second primary cancer	51 [44; 60]	48 [40; 58]	52 [44; 61]	0.02
HIV contamination route *				<0.0001
Heterosexual	129 (29.05)	45 (60.00)	84 (22.76)	
MSM	212 (47.75)	-	212 (57.45)	
IVDU	58 (13.06)	22 (29.33)	36 (9.76)	
Other/unknown	45 (10.14)	8 (10.67)	37 (10.03)	
Nadir CD4/mm^3^				
Nadir CD4 > 500	12 (3.26)	2 (2.86)	10 (3.36)	0.77
200 < Nadir CD4 ≤ 500	73 (19.84)	16 (22.86)	57 (19.13)	
Nadir CD4 < 200	283 (76.90)	52 (74.29)	231 (77.52)	
Period of HIV diagnosis				
Between 1983 and 1989	133 (29.95)	24 (32.00)	109 (29.54)	0.43
Between 1990 and 1995	130 (29.28)	26 (34.67)	104 (28.18)	
Between 1996 and 2001	80 (18.02)	8 (10.67)	72 (19.51)	
Between 2002 and 2007	57 (12.84)	9 (12.00)	48 (13.01)	
After 2007	44 (9.91)	8 (10.67)	36 (9.76)	
HCV antibodies				
Negative	373 (84.01)	50 (66.67)	323 (87.53)	<0.0001
Positive	71 (15.99)	25 (33.33)	46 (12.47)	
HBs antigenemia				
Negative	351 (90.93)	61 (89.71)	290 (91.19)	0.7
Positive	35 (9.07)	7 (10.29)	28 (8.81)	
First cancer occurrence according to ART initiation				
Before	44 (0.91)	8 (10.67)	36 (9.76)	0.81
After	400 (90.09)	67 (89.33)	333 (90.24)	
Time delay between first ART and first cancer (y)	3 (0–9)	6 (3–12)	2 (0–8)	<0.0001
First cancer occurrence according to ART period				
Before ART era (≤1996)	99 (22.30)	9 (12.00)	90 (24.39)	0.02
During ART era (>1996)	345 (77.70)	66 (88.00)	279 (75.61)	
Time delay between first and second primary cancer (y)	4 [1,2,3,4,5,6,7,8,9]	3 [1,2,3,4,5,6,7,8,9,10]	4 [1,2,3,4,5,6,7,8,9]	0.26
Status at time of censored database				0.09
Alive	246 (55.41)	50 (66.67)	196 (53.12)	
Dead	156 (35.14)	19 (25.33)	137 (37.13)	
Lost from follow up	42 (9.46)	6 (8.00)	36 (9.76)	
First primary cancer category				
ADCs	269 (60.59)	32 (42.67)	237 (64.23)	<0.0001
VR-NADCs	51 (11.49)	6 (8.00)	45 (12.20)	
VU-NADCs	124 (27.93)	37 (49.33)	87 (23.58)	
Second primary cancer category				
ADCs	130 (29.28)	15 (20.00)	115 (31.17)	0.14
VR-NADCs	85 (19.14)	15 (20.00)	70 (18.97)	
VU-NADCs	229 (51.58)	45 (60.00)	184 (49.86)	
Tobacco consumption **				
Past	113 (31.48)	12 (20.69)	101 (33.55)	0.12
Current	134 (37.33)	23 (39.66)	111 (36.88)	
Never	112 (31.20)	23 (39.66)	89 (29.57)	
Alcohol consumption ***				
Past	29 (9.83)	4 (8.00)	25 (10.20)	0.002
Current	126 (42.71)	11 (22.00)	115 (46.94)	
Never	140 (47.46)	35 (70.00)	105 (42.86)	

Abbreviations: * SM: men who have sex with men; IDVU: intravenous drug use; ** available in 359 patients, 81.6% of men and 77.3% of women; *** available in 295 patients, 66.4% of men and 66.6% of women.

**Table 2 cancers-14-00401-t002:** Second primary cancer types after a first primary cancer that was potentially related to HPV in men and women living with HIV in the French Dat’AIDS cohort.

Primary Cancers Potentially Related to HPV	*n* = 34	Second Primary Cancer Types	N
Men (*n* = 369)	22 (5.9)		
Oral cavity/pharynx cancers	9		
C02: tongue, other location not specified cancer	4	C01: base of the tongue	3
		C34: lung and bronchial	1
C09: tonsil cancer	4	C02: tongue, other location not specified	1
		C22: liver and intrahepatic bile duct	1
		C34: lung and bronchial	1
		C67: urinary bladder	1
C14: lip, oral cavity and pharynx locations poor specified cancer	1	C85: other lymphoma not specified	1
Anal cancer	12	C02: tongue, other location not specified	1
		C03: gum	1
		C09: tonsil	1
		C18: colon	1
		C20: rectum	1
		C43: melanoma	1
		C44: skin carcinoma	1
		C46: Kaposi sarcoma	1
		C50: breast	1
		C61: prostate	1
		C81: Hodgkin lymphoma	1
		C84: NK/T cell Lymphoma	1
Penis cancer	1	C18: colon	1
Women (*n* = 75)	12 (16.0)		
C21: anal cancer	2	C19: rectosigmoid junction	1
		C49: sarcoma/connective tissu	1
C51: vulva cancer	2	C21: anus	1
		C50: breast	1
C53: cervical cancer	8	C44: skin carcinoma	3
		C21: anus	1
		C22: liver and intrahepatic bile duct	1
		C46: Kaposi sarcoma	1
		C49: sarcoma/connective tissue	1
		C50: breast	1

## Data Availability

The data is available for request from the corresponding author.

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
