# Peer review of "Prevalence and Spectrum of Second Primary Malignancies among People Living with HIV in the French Dat’AIDS Cohort"

_cancers, 2022, doi:10.3390/cancers14020401_

Round 1

Reviewer 1 Report

With the aging PLWHIV population and longer life expectancies, it is obvious that this population will encounter higher risks of developing a second primary cancer. This manuscript is interesting in that it looks into the spectrum and prevalence of SPCs stratified by the first primary cancer in PLWHIV. Below are some comments:

  1. There was a lot of data that was presented but none were particularly interesting. This was primarily because nothing that was really focused on or significant. It would be more interesting if more statistics were fun to see what contribution that sex, nadir CD4, etc has on SPCs. Authors can possibly look into statistical models to do this.
  2. Also the authors claim that PLWHIV SPC risk of 9.1% emphasizes the need for cancer prevention; however, how does this compare to the general population? The authors report an SPC incidence rate of 7.3% for the general population but this cannot be compared to prevalence rates. I think this paper would be strengthened if the same comparisons were done in the general population (as a control) to determine the contribution of HIV in SPC risk. 
  3. The authors report HPV-related cancers and these cancers could possibly be expanded on because it is known that HIV affects HPV-clearance that leads to HPV-related cancer risks, particularly among among MSMs. 

Overall, the authors have a lot of data but in its current analyses, do not show much significance. However, the data can be expanded on to be more interesting. By knowing the true differences from the general population, it will make the data more relevant and provide more insight into SPCs in PLWHIV and direct screening guidelines. 

Author Response

Reviewer 1

With the aging PLWHIV population and longer life expectancies, it is obvious that this population will encounter higher risks of developing a second primary cancer. This manuscript is interesting in that it looks into the spectrum and prevalence of SPCs stratified by the first primary cancer in PLWHIV. Below are some comments:

  1. There was a lot of data that was presented but none were particularly interesting. This was primarily because nothing that was really focused on or significant. It would be more interesting if more statistics were fun to see what contribution that sex, nadir CD4, etc has on SPCs. Authors can possibly look into statistical models to do this.

Answer:

We thank the reviewer for this comment. However, due to the lack of data, such as treatment of the first cancer, HIV viral load, CD4 cell count at each cancer event, we were unable to perform such a study. However, in this study, prevalence and spectrum of SPCs were analyzed according to sex and most subjects, both men and women, had a nadir CD4<200/mm3.

The manuscript has been modified in the section 2. Materials and methods

Line 123: “HIV viral load (VL) and  CD4 count at the time of each cancer event and at the time of HIV-diagnosis were not available as was treatment of the first cancer”

  1. Also the authors claim that PLWHIV SPC risk of 9.1% emphasizes the need for cancer prevention; however, how does this compare to the general population?

Answer: The reviewer is right. Cancer prevention is also relevant to cancer survivors in the general population.

The abstract has been modified as follow in the conclusion:

Conclusion: The SPC prevalence of 9.1% emphasizes the need to strengthen cancer prevention in HIV- positive cancer survivors. Given the aging of the population of PLWH, clinicians have to consider the risk of developing subsequent cancers among HIV-positive cancer survivors of which the frequency and pattern differ according to the first primary cancer type and sex.

The manuscript has been modified as follow in accordance with the request of another reviewer in the Paragraph 4. Discussion

Line 323:Among women of our cohort, breast cancer was the most common SPC after a primary AIDS- defining NHL, in agreement with data of Mahal et al24. Additionally, as observed in the general population of French women27, breast cancer was also the most common SPC after a primary breast cancer. This result is all the more worrying that in France, breast cancer screening has been reported to be lower in WLWH than in the general population (82.2% versus 88%, respectively). These data highlight the need to incentivize WLWH who have survived cancer to screen for breast cancer

An we added a reference 

28 : Tron L, Lert F, Spire B, Dray-Spira R, Agence Nationale de Recherche sur le Sida et les Hépatites Virales (ANRS)‐Vespa2 Study Group. Levels and determinants of breast and cervical cancer screening uptake in HIV-infected women compared with the general population in France. HIV Med. 2017;18(3):181‑95.

Line 341: We observed that after primary cancer of the upper aerodigestive tract in men, cancers of the oral cavity and pharynx represented 41% of the SPCs, and lung cancer, 17%. High levels of tobacco and alcohol exposures were related to the reciprocal associations observed between cancers of the head and neck, esophagus, larynx and lung in France5 as in other countries6,29–31. Among men of our cohort, 70.4% were past or current smokers and 57.1%, past or current alcohol consumers: after primary cancer of the upper aerodigestive tract, cancers of the oral cavity and pharynx represented 41% of the SPCs, and lung cancer, 17%. These data underscore the need to strengthen smoking and alcohol risk reduction among HIV-positive cancer survivors.

And in Paragraph 5.Conclusion, Line 365

In conclusion, given the aging of the population of PLWH, clinicians have to consider the risk of developing subsequent cancers among HIV-positive cancer survivors. The SPC prevalence of 9.1% emphasizes the need to strengthen cancer prevention in HIV-positive cancer survivors and to reinforce smoking and alcohol risk reduction. The two most common SPCs were NHL in men and breast cancer in women, but clinicians must be aware that the frequency and pattern of SPCs differ according to the first primary cancer type and sex. Further studies are needed to determine the excess risk of each SPC type according to the first primary cancer type in PLWH, and the need for more appropriate screening procedures in this population34–36.

  1. The authors report an SPC incidence rate of 7.3% for the general population but this cannot be compared to prevalence rates.

Answer: The reviewer is right. We apologized for this mistake.

The manuscript has been modified as follow paragraph 4. Discussion

Line 249: In the general French population, data from several cancer registries covering 289 967 patients with a first cancer diagnosed between 1989 and 2004 and follow-up until December 31, 2007 indicated an SPC incidence rate of 7.3% of patients who developed a SPC.

  1. I think this paper would be strengthened if the same comparisons were done in the general population (as a control) to determine the contribution of HIV in SPC risk. 

Answer: We agree with the reviewer but it was not possible to carry out such a study

  1. The authors report HPV-related cancers and these cancers could possibly be expanded on because it is known that HIV affects HPV-clearance that leads to HPV-related cancer risks, particularly among among MSMs. 

Answer: We thanks the reviewer for this remark. The manuscript has been modified as follow in the paragraph 4.discussion

Line 338: In addition, long-term persistence of HPV, especially high-risk HPV, is more common among HIV-positive men and women and may contribute to the development of HPV-related SPCs in PLWH.

And we added two references:

  1. Sally N Adebamowo 1 2   3 , Oluwatoyosi Olawande  4 , Ayotunde Famooto  3   4 , Eileen O Dareng  5 , Richard Offiong  6 , Clement A Adebamowo  1   2   3   7 , H3Africa ACCME Research GroupPersistent Low-Risk and High-Risk Human Papillomavirus Infections of the Uterine Cervix in HIV-Negative and HIV-Positive Women Front Public Health . 2017 Jul 21;5:178.  PMID: 28785554
  2. Mary K Grabowski 1 , Ronald H Gray 2 , David Serwadda  3 , Godfrey Kigozi  4 , Patti E Gravitt  5 , Fred Nalugoda  4 , Steven J Reynolds  6 , Maria J Wawer  2 , Stephen Watya  7 , Thomas C Quinn  8 , Aaron A R Tobian  9  High-risk human papillomavirus viral load and persistence among heterosexual HIV-negative and HIV-positive men Sex Transm Infect . 2014 Jun;90(4):337-43. PMID: 24482488
  3. Overall, the authors have a lot of data but in its current analyses, do not show much significance. However, the data can be expanded on to be more interesting. By knowing the true differences from the general population, it will make the data more relevant and provide more insight into SPCs in PLWHIV and direct screening guidelines. 

Answer: We agree with the reviewer as said above. In the conclusion, we stress the need for further studies to determine the excess risk of each SPC type according to the first primary cancer type in PLWH, and the need for more appropriate screening procedures in this population.

Reviewer 2 Report

Authors of this study aimed at investigating the prevalence and spectrum of the second primary cancers among people living with HIV in a French Dat’AIDS cohort. This is an import study highlighting a significant problem that warrants public attention. Overall, the study design and methods are well conducted and results well presented.

Here are a few concerns/suggestions to be addressed.

  1. A sentence on line 134 should start with "Our of 444 patients with SPC..... 
  2. The major concern is the way the discussion section was written. It is not coherent enough, difficult to follow what is being discussed. It repeats the results section for most part without really discussing issues and trying to propose/speculate possible reasons for the findings. This section needs to be rewritten for most part in order to clearly discuss the issues. Linkage to population data needs to be clear.
  3. The authors highlighted in the summary and introduction that the prevalence of primary cancers is at an increase in PLWH, however, results of this study suggests that the prevalence of SPC is similar to that of general population, can they give reasons for this finding or at least speculate? 
  4. ART is known to improve lifespan of PLWH and hence an increased risk for cancers. Similarly, phenomenon like inflammaging and probably suboptimal immune responses are thought to increase cancer risk, any possible reasons as to why prevalence of SPC in PLWH is similar to that of general population?
  5. The conclusion suggests that emphasis should be made to strengthen cancer prevention in PLWH, while this is important, it is not supported by the findings of this study. There is not significant difference in prevalence of SPC between PLWH and the general population as presented by the authors.

Author Response

Reviewer 2

Authors of this study aimed at investigating the prevalence and spectrum of the second primary cancers among people living with HIV in a French Dat’AIDS cohort. This is an import study highlighting a significant problem that warrants public attention. Overall, the study design and methods are well conducted and results well presented.

Here are a few concerns/suggestions to be addressed.

  1. A sentence on line 134 should start with "Our of 444 patients with SPC..... 

Answer: We thanks the reviewer for this comment.

The manuscript has been modified  in the paragraph 3.1 Patients, as follow:

Line 137 There Our of 444 patients with SPCs were 397 (89.4%) patients with one SPC, 43 (9.7%) with two SPCs and 4 (0.9%) with three SPCs.

  1. The major concern is the way the discussion section was written. It is not coherent enough, difficult to follow what is being discussed. It repeats the results section for most part without really discussing issues and trying to propose/speculate possible reasons for the findings. This section needs to be rewritten for most part in order to clearly discuss the issues. Linkage to population data needs to be clear.

Answer : We thanks the reviewer for this comment. The discussion has been rewritten as follow from Line 253 to 349

One would have expected a higher prevalence of SPCs compared to the general population due to the high number of risk factors for cancer in PLWH. The median age of our cohort, which only includes 6.53% of subjects over 65, the prognosis more severe before ART era for PLWH with cancer (the first cancer occurred in the ART era for 77.7% of our patients) could explain this result.

This The rate of prevalence observed in our study is also consistent with the study on HIV-positive San Francisco residents conducted by Hessol et al, between 1985 and 2013 in which SPC accounted for 9% of all cancers12 but is lower than reported by the French CANCERVIH network evaluated on a more recent period13. In this study, 13% of PLWH with a cancer diagnosed between 2014 and 2019 had a history of at least one cancer. This discrepancy might be explained by the fact that this study was conducted on a more recent period and suggests that prevalence of SPCS among PLWH is increasing. However, Ffurther studies are needed to investigate whether prevalence of SPCs among PLWH is increasing confirm such evolution.

Regarding patients’ characteristics, The median delay before the occurrence of SPCs was 4 years but 25% of SPCs were diagnosed within one year of the first primary cancer and 75 within 9 years. as Iin the general population15, SPCs may also occur a few months or many years after the first cancer. We reported but at younger median ages at the first primary cancer event and SPC event than in the general French population5,16,17, explained by due to the differences between both populations of the age distribution18.

Most subjects, both men and women, had a nadir CD4 <200/mm3, but the proportion of those contaminated with HIV through IVDU and with HCV coinfection was higher in women than in men, potentially explaining younger ages in women at the first primary cancer19. as well as the difference in The distribution of the first primary cancer categories (p<.0001) but not of the SPCs categories differed by sex, with a higher percentage of VU-NADCs among women. Interestingly, the distribution of SPC categories was similar in men and women, VU-NADCs being the most common SPCs. This result could not be further explored in this study. A higher prevalence of VR-NADCs might have been expected due to the high prevalence of coinfection with oncogenic viruses in PLWH. Data regarding co-infection with oncogenic viruses in our cohort were limited, making us unable to further explore this result.  

Additionally, we observed that the delay between HIV diagnosis, first ART initiation and the first primary cancer were significantly longer in women than in men. The higher barriers to access and engagement in HIV care for women living with HIV might explained these disparities20–22.

Regarding the pattern of first primary cancers in our cohort, the two most common first primary cancer were KS and NHL in men and NHL and breast cancer in women.   In the general French population of cancer survivors, an excess risk of SPCs was identified after a first cancer of the head and neck, larynx, esophagus, bladder, lip, kidney, lung or chronic lymphatic leukemia in men and after a first cancer of the larynx, head and neck, esophagus, vagina, vulva or acute myeloid leukemia in women5. In our cohort, the two most common first primary cancer were KS and NHL in men and NHL and breast cancer in women. Likewise, the spectrum of SPCs observed in our cohort also differs from that observed in the general French population5 but is consistent with the spectrum of primary cancers in the cohort, as reported12,13,23,24.

Furthermore, Tthe frequency and pattern of SPCs differed according to the first primary cancer, as reported by Hessol et al12, but also by sex: in men, NHL was the most common SPC after primary KS, and KS was the most common SPC after primary NHL, while in women, breast cancer was the most common SPC after primary NHL and primary breast cancer.

After KS,A a higher risk of developing NHL after KS among PLWH than among the general U.S. population was identified during the pre-ART era25, remaining 13-fold higher in the ART era23. Additionally, in this last study, the risk of developing SPC after KS was 2.87 times higher than that in the general U.S. population, with higher risks for NHL, HL, and anal, penile, tongue, and liver cancers. Another study reported a same trend for the overall incidence of SPCs after KS, decreasing from 3.36 in pre-ART era to 1.94 in ART-era26, but the persistence of an increasing risk of developing anal and liver cancer, HL, and NHL, and a significant association with cancers of the tongue and penis and acute lymphocytic leukemia. The spectrum of SPCs after KS observed among men in our cohort was in line with these data.

A higher risk of subsequent non-lymphoid cancers was also recently identified among PLWH after an AIDS-defining and non-AIDS-defining lymphoid malignancy24. In this study, the risk of KS, oral cavity, colon, anus and miscellaneous cancers and myeloid malignancies was significantly increased following an AIDS-defining lymphoid malignancy, whereas the risk of rectal, anus and female breast cancers and myeloid malignancies was significantly increased after non- AIDS- defining lymphoid malignancy24. The pattern of SPCs following HL observed in our cohort of men was in line with these data.

Among women of our cohort, breast cancer was the most common SPC after a primary AIDS- defining NHL, in agreement with data of Mahal et al24. Additionally, as observed in the general population of French women27, breast cancer was also the most common SPC after a primary breast cancer. These results are all the more worrying that in France, breast cancer screening has been reported to be lower in WLWH than in the general population (82.2% versus 88%, respectively) (ref). These data highlight the need to further encourage WLWH to be screened for breast cancer.

We have added this reference :

Tron L, Lert F, Spire B, Dray-Spira R, Agence Nationale de Recherche sur le Sida et les Hépatites Virales (ANRS)‐Vespa2 Study Group. Levels and determinants of breast and cervical cancer screening uptake in HIV-infected women compared with the general population in France. HIV Med. 2017;18(3):181‑95.)

Furthermore, A as reported in general population, we observed a clustering of genital cancers in women with first primary cancers of the breast or the female reproductive tract8.

Additionally, we found that 23.5% of SPCs that occurred after a first primary cancer potentially related to HPV in men and women were also potentially related to HPV. This result is consistent with data from the general French population wherein an increased risk of new malignancies was identified after a potentially HPV-related first cancer, with the second cancer occurring frequently at other HPV-related sites, in men and women28. In addition, long-term persistence of HPV, especially high-risk HPV, is more common in HIV-positive men and women, and may contribute to the development of HPV-related SPCs in PLWH.

We have added these two references

  1. Sally N Adebamowo 1 2   3 , Oluwatoyosi Olawande  4 , Ayotunde Famooto  3   4 , Eileen O Dareng  5 , Richard Offiong  6 , Clement A Adebamowo  1   2   3   7 , H3Africa ACCME Research GroupPersistent Low-Risk and High-Risk Human Papillomavirus Infections of the Uterine Cervix in HIV-Negative and HIV-Positive Women Front Public Health . 2017 Jul 21;5:178.  PMID: 28785554
  2. Mary K Grabowski 1 , Ronald H Gray 2 , David Serwadda  3 , Godfrey Kigozi  4 , Patti E Gravitt  5 , Fred Nalugoda  4 , Steven J Reynolds  6 , Maria J Wawer  2 , Stephen Watya  7 , Thomas C Quinn  8 , Aaron A R Tobian  9  High-risk human papillomavirus viral load and persistence among heterosexual HIV-negative and HIV-positive men Sex Transm Infect . 2014 Jun;90(4):337-43. PMID: 24482488

We observed that after primary cancer of the upper aerodigestive tract in men, cancers of the oral cavity and pharynx represented 41% of the SPCs, and lung cancer, 17%. High levels of tobacco and alcohol exposures were related to the reciprocal associations observed between cancers of the head and neck, esophagus, larynx and lung in France5 as in other countries6,29–31. Among men of our cohort, 70.4% were past or current smokers and 57.1%, past or current alcohol consumers: after primary cancer of the upper aerodigestive tract, cancers of the oral cavity and pharynx represented 41% of the SPCs, and lung cancer, 17%. These data underscore the need to strengthen smoking and alcohol risk reduction among HIV-positive survivors.

  1. The authors highlighted in the summary and introduction that the prevalence of primary cancers is at an increase in PLWH, however, results of this study suggests that the prevalence of SPC is similar to that of general population, can they give reasons for this finding or at least speculate? 

Answer: We thank the reviewer for this remark. The manuscript has been modified as follow in the section 4.discussion

Line 258: This The rate of prevalence observed in our study is also consistent with the study on HIV-positive San Francisco residents conducted by Hessol et al, between 1985 and 2013 in which SPC  accounted for 9% of all cancers12 but is lower than reported by the French CANCERVIH network13 evaluated on a more recent period. In this study, 13% of PLWH with a cancer diagnosed between 2014 to 2019 had a history of at least one cancer. This discrepancy might be explained by the fact that this study was conducted on a more recent period and suggests that prevalence of SPCs among PLWH is increasing. However, Ffurther studies are needed to investigated weither prevalence of SPCs among PLWH is increasing confirm such evolution.

  1. ART is known to improve lifespan of PLWH and hence an increased risk for cancers. Similarly, phenomenon like inflammaging and probably suboptimal immune responses are thought to increase cancer risk, any possible reasons as to why prevalence of SPC in PLWH is similar to that of general population?

Answer: We thank the reviewer for this comment.

The manuscript has been modified as follow line 253 in the section 4.discussion

Line 253:One would have expected a higher prevalence of SPCs compared to the general population due to the high number of risk factors for cancer in PLWH. The median age of our cohort, which only includes 6.53% of subjects over 65, the prognosis more severe before ART era for PLWH with cancer (the first cancer occurred in the ART era for 77.7% of our patients) could explain this result. 

  1. The conclusion suggests that emphasis should be made to strengthen cancer prevention in PLWH, while this is important, it is not supported by the findings of this study. There is not significant difference in prevalence of SPC between PLWH and the general population as presented by the authors.

Answer: The reviewer is right. 

The manuscript has been modified as follow in the paragraph 5. Conclusion

Line 364:In conclusion, given the aging of the population of PLWH, clinicians have to consider the risk of developing subsequent cancers among HIV-positive cancer survivors. The SPC prevalence of 9.1% emphasizes the need to strengthen cancer prevention in HIV-positive cancer survivors and to reinforce smoking and alcohol risk reduction. The two most common SPCs were NHL in men and breast cancer in women, but clinicians must be aware that the frequency and pattern of SPCs differ according to the first primary cancer type and sex. Further studies are needed to determine the excess risk of each SPC type according to the first primary cancer type in PLWH, and the need for more appropriate screening procedures in this population34–36.

And we have added in the paragraph 4. Discussion Line 323

Line 323: Among women of our cohort, breast cancer was the most common SPC after a primary AIDS- defining NHL, in agreement with data of Mahal et al24. Additionally, as observed in the general population of French women27, breast cancer was also the most common SPC after a primary breast cancer. These results are all the more worrying that in France, breast cancer screening has been reported to be lower in women living with HIV(WLWH) than in the general population (82.2% versus 88%, respectively) (28). These data highlight the need to further encourage WLWH to be screened for breast cancer

We have added one reference :

 28 .Tron L, Lert F, Spire B, Dray-Spira R, Agence Nationale de Recherche sur le Sida et les Hépatites Virales (ANRS)‐Vespa2 Study Group. Levels and determinants of breast and cervical cancer screening uptake in HIV-infected women compared with the general population in France. HIV Med. 2017;18(3):181‑95.)

And in Line 341: We observed that after primary cancer of the upper aerodigestive tract in men, cancers of the oral cavity and pharynx represented 41% of the SPCs, and lung cancer, 17%. High levels of tobacco and alcohol exposures were related to the reciprocal associations observed between cancers of the head and neck, esophagus, larynx and lung in France5 as in other countries6,29–31. Among men of our cohort, 70.4% were past or current smokers and 57.1%, past or current alcohol consumers: after primary cancer of the upper aerodigestive tract, cancers of the oral cavity and pharynx represented 41% of the SPCs, and lung cancer, 17%. These data underscore the need to strengthen smoking and alcohol risk reduction among HIV-positive survivors.

Reviewer 3 Report

The findings presented in the study is consistent with several studies conducted in different population of people living with HIV/AIDS (PLWHA). This study particularly indicating the higher risk of second primary malignancies among PLWHA in French Dat’ AIDS cohort, imparts further emphasis on earlier and/or more intensive cancer screening on PLWHA who survived primary cancers.

Comments:

  1. Previous studies suggest that with the use of antiviral therapy, the incidence of first and second primary AIDS-defining cancer seemed to have declined while the incidence of second primary non-AIDS defining cancers has increased. Is this trend similar in the current study? Could chemotherapy, radiation or oncogenic viruses be risk factors for increased second primary non-AIDS defining cancers?
  2. As PLWHAs have higher risk of coinfection with oncogenic viruses and these viruses progress more rapidly in these population, there seem to a greater risk of virus-related second primary cancers. Which type of second primary cancers are more frequent in the population being studied: VR-ADC or VU-ADC?
  3. Legends missing in the y-axis of figure 1 and 2.

Author Response

Reviewer 3

The findings presented in the study is consistent with several studies conducted in different population of people living with HIV/AIDS (PLWHA). This study particularly indicating the higher risk of second primary malignancies among PLWHA in French Dat’ AIDS cohort, imparts further emphasis on earlier and/or more intensive cancer screening on PLWHA who survived primary cancers.

Comments:

  1. Previous studies suggest that with the use of antiviral therapy, the incidence of first and second primary AIDS-defining cancer seemed to have declined while the incidence of second primary non-AIDS defining cancers has increased. Is this trend similar in the current study?

Answer: We thank the reviewer for these comments. This trend could not be assessed in our study as incidence of SPC has not been calculated.

The manuscript has been modified as follow in accordance with the request of another reviewer in the section 4.Discussion

Line 258: The rate of prevalence observed in our study is also consistent with the study on HIV-positive San Francisco residents conducted by Hessol et al,between 1985 and 2013, in which SPC accounted for 9% of all cancers12 but is lower than reported by the French CANCERVIH network13. In this study, 13% of PLWH with a cancer diagnosed between 2014 to 2019 had a history of at least one cancer. This discrepancy might be explained by the fact that this study was conducted on a more recent period and suggest that prevalence of SPCs among PLWH is increasing. However, further studies are needed to confirm such evolution.

  1. Could chemotherapy, radiation or oncogenic viruses be risk factors for increased second primary non-AIDS defining cancers?

Answer: We thanks the reviewer for this question. All these factors may contribute to SPCs in PLWH as we explained in the section 1. Introduction at the line 70, for the general population. Unfortunately in our study, the treatment of first primary cancer was not available making us unable to analyse associated factors with SPC occurrence in PLWH.

 The manuscript has been modified in the section 2.Material and methods Line 123

“HIV viral load (VL),  CD4 count at the time of each cancer event and at the time of HIV-diagnosis were not available as was treatment of the first cancer”

  1. As PLWHAs have higher risk of coinfection with oncogenic viruses and these viruses progress more rapidly in these population, there seem to a greater risk of virus-related second primary cancers. Which type of second primary cancers are more frequent in the population being studied: VR-ADC or VU-ADC?

Answer: We thanks the reviewer for this comment. One would have expected a higher proportion of VR-NADCs among SPCs while VU-NADCs were the most common SPCs in this study.

The manuscript has been modified in the section 4. Discussion Line 276

As follow

The distribution of the first primary cancer categories but not of the SPCs differed by sex, with VU-NADCs being the most common SPCs. This result could not be further explored in this study.” A higher prevalence of VR-NADCs might have been expected due to the high prevalence of coinfection with oncogenic viruses in PLWH. Data regarding co-infection with oncogenic viruses in our cohort were limited, making us unable to further explore this result.

  1. Legends missing in the y-axis of figure 1 and 2.

Answer: We apologize for this mistake. The legends have been added

We have also added the meaning of the abbreviations under tables 1 and 2

Round 2

Reviewer 1 Report

Although the knowledge gained from this study would be useful, the research design does not strongly support any results or conclusions that are made. The authors made a good attempt to address previous comments by adding in more references and adjusting portions of the text but the weakness is really in the design and the fact that there was really no significance in the data. The data was purely descriptive and thus this paper may be better suited for submission as a letter to the editor. Overall, the topic is interesting but the conclusion (that the risk for developing subsequent cancers) can be assumed given the aging population of PLWH, even without presenting the data. Further, some additional text has been added in which the reasoning is not entirely clear (page 16, lines 253-257 and page 17, lines 263-266). 

Author Response

As suggested by the reviewer, in order to avoid confusion, we have modified part of the discussion (lines 253-257) and (lines 263-266) as follows:

Line 253-257: “One would have expected a higher prevalence of SPCs compared to the general population. The more severe prognosis in the pre -ART era for PLWH with cancer and the low percentage of subjects over 65 years of age in our cohort could explain the lower prevalence rate of SPCs observed in our cohort than expected, due to the high number of risk factors for cancer in PLWH.

The median age of our cohort, which only includes 6.53% of subjects are over 65. Moreover, the first cancer occurred in the ART era for 77.7% of our patients, suggesting that most of patients the prognosis more severe before ART era for PLWH with cancer (

, ) could explain this result.

Line 268 : “This discrepancy might be explained by the fact that this study was conducted on a more recent period and suggests that prevalence of SPCs among PLWH is increasing. However,. Further studies are needed to confirm such evolution.”

The conclusion has been modified as follow:

Line 371: ”In conclusion, given the aging of the population of PLWH, clinicians have to consider the risk of developing subsequent cancers among HIV-positive cancer survivors. This descriptive analysis of the spectrum of SPCs in HIV positive cancer survivors identified NHL in men and breast cancer in women as the two most common SPCs. However, clinicians must be aware that the frequency and pattern of SPCs differ according to the first primary cancer type and might change over time, as reported by Mukthar et al for SPCs associated with KS (ref Mukhtar), arguing for a regular epidemiologic monitoring. Further studies are needed to determine the excess risk of each SPC type according to the first primary cancer type in PLWH, and the need for more appropriate screening procedures in this population37–39.

The conclusion of the abstract was modified as follow:

“Given the aging of the population of PLWH, clinicians have to consider the risk of developing of which The frequency and pattern of subsequent cancers among HIV-positive cancer survivors differ according to the first primary cancer type and sex”.